# Benchmarks for Corruption Invariant Person Re-identification

**Minghui Chen**[*]   **Zhiqiang Wang**[*]   **Feng Zheng**[†]
Department of Computer Science and Engineering
Southern University of Science and Technology
Shenzhen 518055, P.R. China
`{ming_hui.chen, wangzq_2021}@outlook.com`   `zfeng02@gmail.com`

## Abstract

When deploying person re-identification (ReID) model in safety-critical applications, it is pivotal to understanding the robustness of the model against a diverse array of image corruptions. However, current evaluations of person ReID only consider the performance on clean datasets and ignore images in various corrupted scenarios. In this work, we comprehensively establish five ReID benchmarks for learning corruption invariant representation. In the field of ReID, we are the first to conduct an exhaustive study on corruption invariant learning in single- and cross-modality datasets, including Market-1501, CUHK03, MSMT17, RegDB, SYSU-MM01. After reproducing and examining the robustness performance of 21 recent ReID methods, we have some observations: 1) transformer-based models are more robust towards corrupted images, compared with CNN-based models, 2) increasing the probability of random erasing (a commonly used augmentation method) hurts model corruption robustness, 3) cross-dataset generalization improves with corruption robustness increases. By analyzing the above observations, we propose a strong baseline on both single- and cross-modality ReID datasets which achieves improved robustness against diverse corruptions. Our codes are available on `https://github.com/MinghuiChen43/CIL-ReID`.

## 1   Introduction

Person re-identification (ReID) is regarded as a fine-grained instance retrieval problem. Unlike image classification tasks, the goal of person ReID is to match and rank pedestrian images across multiple non-overlapping cameras [50]. Due to its vast applications for intelligent security and video surveillance, person ReID has become a hot topic in computer vision. However, there are still many problems in deploying current person ReID models to the real world. Unlike object classification and detection, ReID is a instance-level recognition and ranking problem that relies on extracting robust and detailed information. Unfortunately, current neural networks are easily confused by various forms of corruptions such as noise, blurring and snow [16]. Therefore, learning invariant representation towards corrupted images in this task is challenging and merits extra investigations.

To comprehensively study the corruption invariant learning of models in various scenarios, we make the first attempt to establish the corruption invariant ReID benchmarks on both single- and cross-modality datasets, including Market-1501 [53], CUHK-03 [23], MSMT17 [46], RegDB [30], and SYSU-MM01 [47]. The statistics of these datasets are shown in Tab. 1. We re-construct these five datasets by applying 20 types of corruption that commonly occur in the real world. Meanwhile,

---

[*]Equal Contribution
[†]Corresponding Author

35th Conference on Neural Information Processing Systems (NeurIPS 2021) Track on Datasets and Benchmarks.

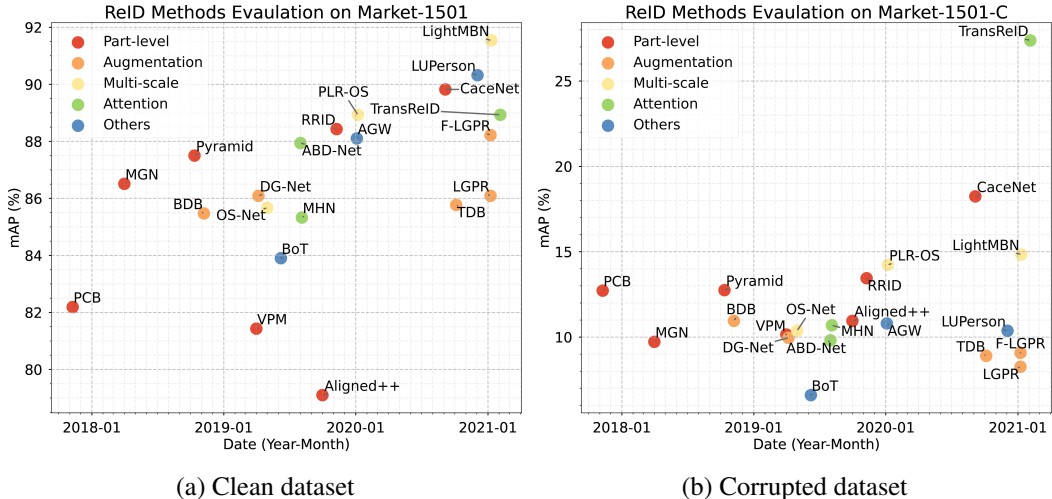

(a) Clean dataset          (b) Corrupted dataset

Figure 1: Large scale evaluation on ReID methods in recent years. (a) The performance of the clean test set (Market-1501) is shown on the left, demonstrating the increasing trend in mAP for various methods over the last few years. (b) The performance of the corrupted test set (corrupted query and gallery) is shown on the right. The overall mAP is significantly lower, and recent methods (e.g. LUPerson) are even less robust than PCB (proposed in 2017). Refer to the appendix for corresponding literature of above methods and performance on other datasets.

each corruption type contains five levels of severity. The benchmark datasets are constructed based on agnostic corruption types that are not encountered in model training. Since corruption types in the real world are numerous and unpredictable, it will be more practical to learn a corruption generalized model without additional training and adaptation. To measure the corruption robustness comprehensively, we present robustness performance in three evaluation settings: 1) corrupted query, 2) corrupted gallery, and 3) corrupted query and gallery.

Based on the corruption invariant learning benchmark, we reproduce 21 advanced ReID methods in recent years and conduct large-scale evaluations on the corruption robustness of various state-of-the-art CNN-based and transformer-based models. While the performance of ReID methods on the clean test set has shown an upward trend in recent years (see Fig. 1), the performance on the corrupted test set is significantly lower, with plenty of room for improvements. Specifically, we have some main findings based on robustness evaluation: 1) transformer-based models [7, 14] excel in corrupted test set comparing with CNN-based models. This demonstrates that the transformer-based models are capable of mining rich structured patterns, which is especially important when dealing with corrupted data. 2) In contrary to the clean test set, increasing the probability of the random erasing [57], a frequently used augmentation technique, impairs model performance on the corrupted test set. We argue that random erasing method hinders models from mining rich discriminative information from corrupted images. 3) Interestingly, the cross-dataset generalization tends to improve with the corruption robustness. This clearly refutes that robustness towards synthetic corruption do not help with robustness on naturally occurring distribution shifts [39].

From the above investigations, we introduce a strong baseline on the corruption invariant learning benchmarks in person ReID. In summary, our contributions are as follows:

- We propose benchmarks for corruption invariant Person ReID, including both single- and cross-modality datasets Market-1501, CUHK-03, MSMT17, RegDB, and SYSU-MM01.

- We reproduce 21 advanced ReID methods and have some interesting findings on learning corruption invariant representation in terms of network architectures and data augmentation.

- We are the first to reveal that cross-dataset generalization tends to increases with corruption robustness. This intriguing finding demonstrates the practical utility of learning a corruption invariant model towards real-world distribution shift, which has been overlooked by previous research on corruption robustness.

- We establish a strong baseline for corruption invariant person ReID, improving random erasing, BNNeck and identity loss.

## 2 Related Work

### 2.1 Person ReID

Existing person ReID techniques fall into two main categories: closed-world and open-world settings. With the performance saturation under closed-world setting, the research focus on person Re-ID has recently shifted to the open-world setting, facing more challenging issues [50]. A feature of the open-world settings is that pedestrian data might be heterogeneous data, including infrared images [30, 47], cross-resolution images [24, 45] and even text descriptions [22]. Corrupted images can be regarded as heterogeneous data for clean images, and our corruption-invariant ReID benchmarks as an open-world ReID setting defined in [50]. Current person ReID system contains three main components: feature representation learning, deep metric learning and ranking optimization [50]. The goal of feature representation learning is to efficiently extract discriminative features. This is accomplished through the use of attention mechanisms [3, 2, 4], the capture of multi-scale features [58, 19, 48], and the mining of local features [38, 42, 52, 32, 37, 27, 51]. In ReID, deep metric learning entails developing a reasonable loss function [54, 18] and devising an appropriate sampling strategy [25]. The purpose of rank optimization is to improve retrieval performance during the inference stage. The most frequently used strategy is to optimize the ranking list by leveraging gallery-to-gallery similarity [56].

### 2.2 Corruption Robustness

The human visual system is not easily fooled by a wide range of image corruptions, such as noise, blurring and pixelation or their combination. In contrast, current deep neural networks suffer from severe performance degradation towards corrupted images [16]. Study on corruption robustness has a long history in computer vision [36, 1, 6] and has recently received more attention due to the release of corruption benchmarks for image recognition, such as CIFAR-10-C, CIFAR-100-C and ImageNet-C [16]. Since then, similar benchmarks on common corruption have also been proposed in the field of object detection [29], semantic segmentation [21] and pose estimation [43]. These benchmarks reveal that the generalization ability of advanced models under corrupted input still needs to be further improved [16, 17, 15]. For improving the corruption robustness, various data augmentation techniques have been proposed recently. For example, AugMix [17] utilizes a formulation to mix multiple augmented images and obtains significant improvement on ImageNet-C. Rusak *et al.* [35] design a data augmentation algorithm based on adversarial framework for defending against common corruptions.

## 3 Corruption Invariant ReID Benchmark

### 3.1 Evaluation Metrics

To evaluate the performance of a ReID system, mAP (mean average precision) [53] and CMC-k (cumulative matching characteristics, a.k.a, Rank-k matching accuracy) [44] are two widely used measurements. Besides common used metrics mAP and CMC-k, we also present mINP (mean inverse negative penalty) to evaluate the ability to retrieve the hardest correct match [50]. For a robust Re-ID system, the correct matches should have low rank values. The mINP is represented by

$$\text{mINP} = \frac{1}{n}\sum_i(1 - \text{NP}_i) = \frac{1}{n}\sum_i(1 - \frac{R_i^{hard} - |G_i|}{R_i^{hard}}) = \frac{1}{n}\sum_i \frac{|G_i|}{R_i^{hard}}, \tag{1}$$

where $R_i^{hard}$ indicates the rank position of the hardest match, $|G_i|$ represents the total number of correct matches for query $i$, and NP represents negative penalty. The INP (the highest, the better) is inverse of NP, which is a computationally efficient metric.

### 3.2 Benchmark Datasets

Our robust person ReID benchmarks are composed of five datasets: Market-1501, CUHK-03, MSMT17, RegDB, and SYSU-MM01. The detailed information and statics of these datasets are shown in Tab. 1. We employ 15 image corruptions from ImageNet-C dataset and 4 image corruptions from Extra ImageNet-C [16]. In addition, we introduce a *rain* corruption type, which is a common

Table 1: Statistics of our benchmarking datasets for single- and cross-modality person ReID.

| Dataset | Time | Total ID | Train ID | Test ID | Total image | Train set | Query | Gallery | Cam. |
|---|---|---|---|---|---|---|---|---|---|
| | | | | | *Single-modality datasets* | | | | |
| CUHK03-detected | 2014 | 1,467 | 767 | 700 | 14,097 | 7,365 | 1,400 | 5,332 | 2 |
| Market-1501 | 2015 | 1,501 | 751 | 750 | 36,036 | 12,936 | 3,368 | 19,732 | 6 |
| MSMT17 | 2018 | 4,101 | 1,041 | 3,060 | 126,441 | 32,621 | 11,659 | 82,161 | 15 |
| | | | | | *Cross-modality datasets* | | | | |
| Dataset | Time | Total ID | Train ID | Test ID | Total image | Train set | Query | Gallery | Cam. |
| RegDB | 2017 | 412 | 206 | 206 | 8,240 | 4,120 | 2,060 | 2,060 | - |
| SYSU-MM01 | 2017 | 491 | 395 | 96 | 303,420 | 34,167 | 3,803 | 301 | 6 |

type of weather condition, but it is missed by the original corruption benchmark (for specific parameter settings and implementation, see our appendix). These corruptions consist of *Noise*: Gaussian, shot, impulse, and speckle; *Blur*: defocus, frosted glass, motion, zoom, and Gaussian; *Weather*: snow, frost, fog, brightness, spatter, and rain; *Digital*: contrast, elastic, pixel, JPEG compression, and saturate. Each corruption has five severity levels, resulting in 100 distinct corruptions.

In contrast to object classification [16] and detection [29], the ReID task is an image pair matching problem with a query and gallery as a test set. To assess the robustness on a broad scale, we present three evaluation settings: both the query and gallery are corrupted, the query is corrupted alone, and the gallery is corrupted alone. For the cross-modality datasets RegDB and SYSU-MM01, only RGB images from the gallery are corrupted. Additionally, considering that the size of the ReID test set has a great impact on performance indicators, we do not directly add all corruption types and all severity levels to the test set. We randomly select one corruption type and one severity level from each image in the test set to create the query or gallery. We repeat the preceding evaluation ten times with the same query and gallery size as the clean test set (three times for large scale datasets MSMT17).

### 3.3 Evaluation Models

Our standard backbone is the widely used ResNet50 [12]. We follow a standard training pipeline, which includes initialization with an ImageNet pre-trained model and modification of the dimension of the fully connected layer to $N$ [26]. $N$ is the number of identities in the training set. In the early training epochs, we adopt a learning warmup strategy. Additionally, the transformer-based models [41] (e.g. ViT [7], DeiT [40]) for object ReID are based on TransReID proposed by He *et al.* [14].

### 3.4 A Strong Baseline

Based on our findings from corruption robustness evaluation, we design a new robust baseline for person ReID, which achieves competitive performance on both single- and cross-modality ReID tasks. Our baseline contains the following key components.

**Local-based augmentation.** Random erasing [57] is an augmentation method that randomly selects a rectangle region in an image and erases its pixel with a random value (see Fig. 2). If no special statement is present, we utilize random cropping, horizontal flipping, and 0.5-probabilistic random erasing as default data augmentations. Random erasing yields consistent improvement on various person ReID datasets, but we find that the performance on corrupted datasets decreases with increasing erasing probability (see Fig. 4 right part). Additionally, we observe that an augmentation method called RandomPatch [58] also degrades the corruption robustness. RandomPatch works by first creating a patch pool of randomly extracted image patches and then pasting a random patch from the patch pool onto an input image at a random position. We believe that these two augmentation methods, which heavily occlude images and introduce additional perturbation information, will impair the models' ability to mine salient local information, which is critical for retrieving corrupted images. To compensate for the loss of discriminative information caused by strong erasing, we propose a data augmentation technique called soft random erasing, in which the erased area is not completely replaced with random pixels but retains a proportion of the original pixels, as shown in Fig. 2 (a). To alleviate the strong perturbation introduced by RandomPatch, we propose a mixing-based augmentation technique called self patch mixing (SelfPatch). As illustrated in Fig. 2 (b), SelfPatch

works by randomly cutting a block from the original image and then remixing it with another random position.

**Consistent identity loss.**   The classical identity (ID) loss [55] is computed by the cross-entropy

$$\mathcal{L}_{id} = -\frac{1}{n}\sum\nolimits_{i=1}^{n}\log(p(y_i|x_i)). \tag{2}$$

Given an input image $x_i$ with label $y_i$, the predicted probability of $x_i$ being recognized as class $y_i$ is encoded with a softmax function, represented by $p(y_i|x_i)$. The identity loss is then computed by the cross-entropy, where $n$ represents the number of training samples within each batch [50]. The previous ID loss only calculates the loss of a single augmented sample per image. To enforce model response smoother for different augmented variants, we utilize the Jensen-Shannon divergence among the posterior distribution of the original sample $x_{\text{orig}}$ and its augmented variants [17]. That is, for $p_{\text{orig}} = \hat{p}(y \mid x_{\text{orig}}), p_{\text{aug1}} = \hat{p}(y \mid x_{\text{aug1}}), p_{\text{aug2}} = \hat{p}(y|x_{\text{aug2}})$. The self identity loss can be computed by first obtaining $M = (p_{\text{orig}} + p_{\text{aug1}} + p_{\text{aug2}})/3$ and then computing

$$\mathcal{L}_{cid}(p_{\text{orig}}; p_{\text{aug1}}; p_{\text{aug2}}) = \frac{1}{3}\Big(\text{KL}[p_{\text{orig}}\|M] + \text{KL}[p_{\text{aug1}}\|M] + \text{KL}[p_{\text{aug2}}\|M]\Big). \tag{3}$$

The gain of training with consistent ID loss is obvious when combining with global data augmentation (e.g. AugMix).

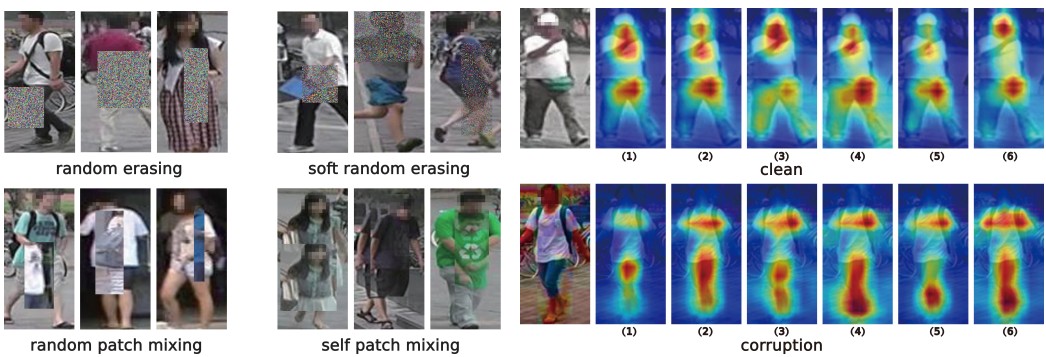

(a) Local-based data augmentation          (b) Visualization of activation map

Figure 2: Visualization of our augmented examples and activation maps. **(a) Left** are four different data augmentation methods. As can be seen, random erasing and random patch mixing introduce severe occlusion compared with soft random erasing and self patch mixing. **(b) Right** are activation maps of models trained with different augmentations. From left to right of each are input images, activation maps from the **(1)** standard model, a model trained with **(2)** AugMix, **(3)** random erasing, **(4)** soft random erasing, **(5)** random patch mixing and **(6)** self patch mixing. Models trained with our proposed augmentations (soft random erasing and random patch mixing) capture more discriminative parts.

**Inference before BNNeck.**   BNNeck [26] is a batch normalization (BN) [20] layer after features for triplet loss and before classifier fully-connected layers in ReID tasks. The motivation of BNNeck is to make features gaussianly distribute near the surface of the hypersphere and make the ID loss easier to converge [26]. However, we discover that the corruption robustness of models will decrease when using features after BNNeck (see appendix). One reasonable explanation for this is that the BN layer will memorize the statistical information of the train set, while the statistical information of the corrupted test set and the clean train set are quite different.

## 4   Experiments

### 4.1   Benchmarking SOTA Methods

In this part, we evaluate the corruption robustness of 21 ReID methods, including AGW [50], BoT [26], ABD-Net [3], OS-Net [58], DG-Net [55], MHN [2], BDB [5], TransReID [14], LGPR [10],

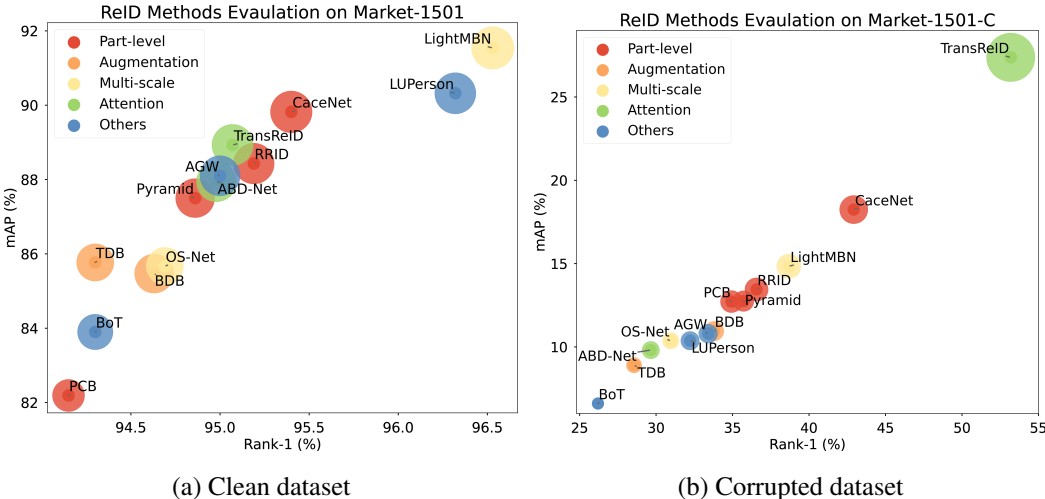

(a) Clean dataset         (b) Corrupted dataset

Figure 3: Performance evaluations of the person ReID methods in recent years. The x-axis, y-axis, and bubble size indicate Rank-1, mAP, and mINP, respectively. (a) Evaluations on clean Market-1501 test dataset. (b) Evaluations on Market-1501-C (corrupted query and gallery). In general, performance on the clean test set is not positively correlated with performance on the corrupted test set, and there is considerable room for improvement on corruption robustness.

F-LGPR [11], TDB [33], LUPerson [8], LightMBN [19], PLR-OSNet [48], CaceNet [51], PCB [38], Pyramid [52], AlignedReID++ [27], RRID [32], VPM [37], and MGN [42] (see our appendix for specific parameter settings of models). Fig. 3 illustrates the Rank-1, mAP, and mINP performance indicators of 21 ReID methods in recent years for both the clean test set and the corrupted dataset (the corrupted query and gallery). The bubble size indicates the relative level of mINP indicator (see the appendix for more details). In general, existing methods perform poorly on the corrupted test set, and there is vast room for improvement. In Fig. 3, there is no obvious trade-off or positive correlation between the model performance on the clean test set and the corrupted test set.

TransReID [14] significantly outperforms other methods in terms of indicators (most notably the mINP) of corrupted test sets. It is worth noting that the mINP index measures the ability to retrieve difficult samples, which makes it an appropriate indicator of the ReID model corruption robustness. From Fig. 1 and 3, we can observe that part-level based ReID methods perform well on clean and corrupted test sets. This demonstrates that learning local features is still critical for the corrupted images, and it can also make the model more robust to corruption variation. On the corrupted test set, the performance of the vanilla PCB [38] is still competitive, even surpassing some methods that perform excellently on the clean dataset.

Some of the above reproduced ReID methods were proposed to learn a noise-robust model. These sample noises include heavy occlusion (e.g. VPM [37]), inaccurate bounding boxes caused by sampling errors (e.g. Pyramid [52]), illumination variation (e.g. BDB [5]), style changing (e.g. DG-Net [55]), and adversarial perturbations (e.g. F-LGPR [11]). But unfortunately, the corruption robustness of the above methods is not particularly strong. Therefore, we argue that the corruption invariant ReID is complementary to the previous research on noise-robust ReID and merits special investigation.

## 4.2 Connection with Generalizable Person ReID

In the previous corruption robustness research, it is less clear how a robust model generalizes across different datasets. For image classification, Taori *et al.* [39] found that current robustness measures for synthetic distribution shift are at most weakly predictive for robustness on the natural distribution shifts presently available. However, we found that corruption robustness measures are predictive for robustness on the natural distribution shifts on person ReID. As illustrated in Fig. 4, extensive experiments with various methods and data enhancements reveal that the ability to generalize across datasets increases as corruption robustness increases. The cross-dataset generalization ability refers to the performance of the model trained on the Market-1501 dataset and tested on another dataset

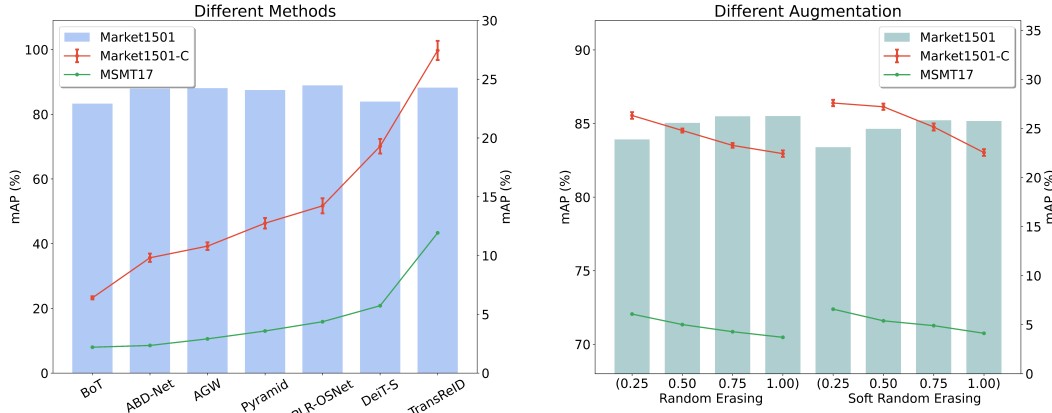

Figure 4: Cross-dataset generalization (Market-1501 to MSMT17) improves as corruption robustness increases. In contrast, the cross-dataset generalization has little correlation with performance on the clean samples. The histogram represents performance on the clean Market-1501 test set, the blue line depicts performance on the Market-1501-C (corrupted query and gallery), and the green line depicts performance when transferring directly to the MSMT17 dataset. On the **left** are the results of various methods, while on the **right** are the results of the same model trained with different augmentations. For different augmentations, the green histograms represent the random (**left** half) and soft random erasing (**right** half), respectively. The value on the x-axis represents the probability of erasure. See our appendix for more experiment results.

(*e.g.* CUHK-03 and MSMT17). The histogram represents the performance on the clean Market-1501, and the mAP is the value on the left y-axis. The blue and green lines respectively reflect the trend of the corruption robustness and the cross-dataset generalization, and the mAP is the value on the right y-axis. As illustrated in the left panel of Fig. 4, the cross-dataset generalization exhibits a consistent upward trend with the corruption robustness (correlation coefficient $\rho = 0.97$). However, there is no obvious correlation between cross-dataset generalization and clean sample performance (correlation coefficient $\rho = 0.22$).

### 4.3 Benchmarking Network Architecture

To further analyze the corruption robustness of the TransReID method, we compare CNN-based models and transformer-based models in this part. The number of parameters (Params) and multi-adds (MACs) of evaluated models are presented. To begin, the performance of TransReID illustrated in Fig. 1 is identical to that of Trans-Vit-base in Fig. 3. Although it outperforms other models, it requires more memory and computation time. Additionally, we present the robustness performance in two settings, one in which only the query is corrupted and one in which only the gallery is corrupted. In general, a corrupted query makes models more difficult to sort simple samples correctly (Rank-1 is low), whereas a corrupted gallery makes models more difficult to retrieve difficult samples (mINP is low). When memory and computational overhead are considered, we discover that the ViT architecture is still superior in terms of corruption robustness. On the corrupted test set (corrupted query and gallery setting), ViT-S and DeiT-S outperform all CNN-based models except for ResNeXt-101-ibn [49]. Additionally, we discover that when ResNet-50 is combined with an IBN [31] module, the corruption robustness of ResNet-50 is significantly improved. This is consistent with their findings [31] that instance normalization (IN) learns features that are invariant to changes in appearance, such as colors and styles. In summary, incorporating attention modules and judicious use of IN into network architecture can significantly improve the corruption robustness.

### 4.4 Benchmarking Data Augmentation

Data augmentation is vital to improve the corruption robustness. From Tab. 3, we find that the AugMix is significantly more effective in boosting robustness than other augmentation methods, which is consistent with previous research [17]. As depicted in the right part of Fig. 4, corruption robust decreases with increasing erasing probability. Besides, soft random erasing are more effective for improving corruption robustness and less sensitive to the tuning of erasing probability compared

Table 2: Comparisons of transformer-based models and CNN-based models. All the models are trained with $256 \times 128$ inputs. Trans- denotes the TransReID method. Compared to the original ViT, the TransReID includes SIE and JPM modules. While ViT-based models generally outperform the CNN-based models in terms of corruption robustness, the advantage of the ViT over CNN-based models with an attention mechanism is not obvious.

| Network | MACs | Params | Clean Eval. | | | Corrupted Eval. | | | Corrupted Query | | | Corrupted Gallery | | |
|---|---|---|---|---|---|---|---|---|---|---|---|---|---|---|
| | (G) | (M) | mINP | mAP | R-1 | mINP | mAP | R-1 | mINP | mAP | R-1 | mINP | mAP | R-1 |
| ResNet-50 | 4.06 | 23.51 | 59.30 | 85.06 | 93.38 | 0.21 | 8.50 | 27.30 | 14.34 | 26.42 | 30.52 | 0.39 | 27.00 | 77.15 |
| ResNet-50-ibn | 4.06 | 23.51 | 67.79 | 86.55 | 94.36 | 0.33 | 13.90 | 37.75 | 20.45 | 36.77 | 43.39 | 0.85 | 35.78 | 83.34 |
| ResNeSt-50 | 4.68 | 25.44 | 65.49 | 87.97 | 95.28 | 0.26 | 9.82 | 30.57 | 18.26 | 31.71 | 37.65 | 0.51 | 31.79 | 82.93 |
| ResNet-101-ibn | 6.49 | 42.50 | 65.27 | 87.90 | 95.22 | 0.37 | 13.96 | 37.35 | 21.99 | 37.35 | 43.42 | 0.90 | 36.61 | 84.09 |
| SE-ResNet-101-ibn | 6.50 | 47.25 | 67.75 | **89.08** | **95.49** | 0.69 | 16.99 | 43.14 | 27.87 | 45.28 | 51.48 | 2.60 | 47.39 | 88.32 |
| ResNeXt-101-ibn | 6.51 | 42.13 | **67.81** | 89.05 | 95.04 | **1.34** | **21.65** | **48.25** | **31.38** | **50.49** | **56.59** | **3.78** | 50.30 | **88.48** |
| DeiT-S | 2.78 | 22.31 | 56.36 | 83.90 | 93.23 | 1.07 | 19.30 | 44.38 | 23.41 | 43.30 | 51.49 | 2.57 | 43.98 | 83.73 |
| ViT-S | 6.16 | 47.81 | 53.34 | 82.42 | 92.79 | 1.02 | 18.06 | 42.47 | 22.76 | 43.94 | 53.39 | 2.68 | 43.64 | 82.83 |
| ViT-S + SIE | 6.16 | 47.81 | 55.99 | 83.70 | 93.35 | 0.82 | 16.88 | 40.26 | 22.54 | 42.34 | 50.76 | 2.29 | 43.07 | 82.96 |
| ViT-S + JPM | 6.94 | 53.72 | 55.42 | 83.40 | 92.70 | 1.10 | 19.33 | 43.62 | 23.85 | 43.84 | 51.91 | 2.55 | 43.87 | 82.86 |
| ViT-B | 11.03 | 85.61 | 64.08 | 87.11 | 94.60 | 1.82 | 25.84 | 51.31 | 31.41 | 51.54 | **59.44** | **4.40** | 52.00 | 87.60 |
| Trans-ViT-S | 11.33 | 53.72 | 57.01 | 84.46 | 93.74 | 1.02 | 18.23 | 42.57 | 23.65 | 43.19 | 50.71 | 2.48 | 43.29 | 83.47 |
| Trans-DeiT-B | 19.55 | 92.70 | 67.16 | 88.54 | 95.07 | 1.70 | 24.71 | 51.67 | 32.39 | 50.79 | 57.12 | 3.84 | 50.03 | 87.65 |
| Trans-ViT-B | 19.55 | 92.70 | **69.31** | **88.97** | **95.10** | **1.93** | **27.10** | **52.77** | **34.52** | **52.30** | 57.96 | 4.18 | **52.19** | **88.60** |

Table 3: Comparisons of various data augmentations. The upper part compares the global augmentation methods (Random affine transformation, AutoAugment and AugMix). The bottom part compares the local-based augmentation methods when combined with the AugMix. REA stands for random erasing, S-REA stands for soft random erasing, R-PATCH stands for random patch mixing, and S-PATCH stands for self patch mixing augmentation.

| Augmentation | Clean Eval. | | | | | Corrupted Eval. | | | | | Corrupted Query | | | | Corrupted Gallery | | | |
|---|---|---|---|---|---|---|---|---|---|---|---|---|---|---|---|---|---|---|
| | mINP | mAP | R-1 | R-5 | R-10 | mINP | mAP | R-1 | R-5 | R-10 | mINP | mAP | R-1 | R-5 | mINP | mAP | R-1 | R-5 |
| Standard | 45.70 | 77.76 | 91.69 | 96.44 | 97.83 | 0.43 | 14.31 | 37.31 | 53.36 | 59.99 | 16.79 | 34.45 | 42.99 | 53.06 | 0.90 | 33.54 | 77.74 | 90.28 |
| R-Affine | 59.34 | 85.69 | 93.88 | 98.16 | 98.93 | 0.27 | 8.01 | 27.46 | 39.70 | 45.20 | 16.28 | 30.69 | 36.34 | 45.04 | 0.83 | 33.02 | 81.29 | 92.12 |
| AutoAug | 46.39 | 80.18 | 92.34 | 97.60 | 98.57 | 0.37 | 10.55 | 30.40 | 42.71 | 48.41 | 14.39 | 31.87 | 40.23 | 49.24 | 1.12 | 35.22 | 80.64 | 91.86 |
| AugMix | 45.92 | 77.16 | 91.03 | 96.88 | 98.13 | 1.05 | 22.47 | 48.06 | 65.07 | 71.27 | 21.79 | 43.46 | 53.93 | 65.58 | 1.95 | 41.32 | 80.25 | 92.21 |
| + REA | 57.10 | 83.40 | 93.08 | 97.83 | 98.43 | 1.49 | 24.32 | 49.80 | 66.82 | 72.68 | 26.86 | 48.08 | 57.33 | 68.92 | 3.30 | 47.46 | 84.71 | 94.45 |
| + S-REA | 57.34 | 83.48 | 92.99 | 97.57 | 98.43 | 2.19 | 26.66 | 52.60 | 69.64 | 75.52 | 28.75 | 50.33 | 56.69 | 71.49 | 4.44 | 49.54 | 85.27 | 94.58 |
| + R-PATCH | 47.37 | 77.78 | 90.97 | 96.53 | 98.10 | 1.00 | 22.05 | 47.49 | 64.35 | 70.42 | 21.97 | 43.20 | 53.60 | 64.84 | 1.74 | 41.19 | 80.85 | 92.45 |
| + S-PATCH | 54.30 | 81.86 | 92.55 | 97.48 | 98.49 | 1.17 | 22.72 | 47.99 | 64.84 | 70.73 | 25.19 | 45.78 | 55.02 | 66.49 | 2.45 | 44.60 | 83.18 | 93.83 |

with random erasing. In Tab. 3, we have a similar observation that soft random benefits more for improving corruption robustness. Meanwhile, the SelfPatch augmentation method outperforms the RandomPatch augmentation method on clean and corrupted test sets.

## 4.5 A Strong Baseline on Corruption Invariant ReID

On the basis of the foregoing investigation, we propose three general and simple techniques for enhancing corruption robustness (detailed ablations see the appendix). The first is the consistent ID loss that enforces a smoother network response [17]. The second technique is inference with features before BNNeck, in case the feature is too domain-specific. The third one is the proposed local-based augmentation techniques, soft random erasing and self patch mixing. Our baseline is CIL (**C**onsistent identity loss, **I**nference before BNNeck and **L**ocal-based augmentation). In single-modality datasets (see Tab. 4), our proposed baseline CIL achieves competitive performance on the clean test set and outstanding results on three corrupted situations. We also evaluate the corruption robustness of the

Table 4: Corruption invariant person ReID benchmarks on single-modality datasets. SBS [13] represents a stronger baseline on top of BoT. In single-modality datasets, our proposed baseline CIL achieves competitive performance on the clean test set and remarkable results on three corrupted scenarios.

| Dataset | Method | Clean Eval. | | | | Corrupted Eval. | | | | Corrupted Query | | | | Corrupted Gallery | | | |
|---|---|---|---|---|---|---|---|---|---|---|---|---|---|---|---|---|---|
| | | mINP | mAP | R-1 | R-5 | mINP | mAP | R-1 | R-5 | mINP | mAP | R-1 | R-5 | mINP | mAP | R-1 | R-5 |
| Market-1501 | BoT | 59.30 | 85.06 | 93.38 | 97.71 | 0.20 | 8.42 | 27.05 | 40.28 | 14.56 | 26.89 | 31.92 | 40.24 | 0.39 | 26.82 | 76.78 | 89.57 |
| | AGW | **64.03** | 86.51 | 94.00 | 98.01 | 0.35 | 12.13 | 31.90 | 46.54 | 19.44 | 31.75 | 35.25 | 44.09 | 0.67 | 33.38 | 80.45 | 91.90 |
| | SBS | 60.03 | **88.33** | **95.90** | **98.49** | 0.29 | 11.54 | 34.13 | 47.28 | 18.47 | 35.33 | 42.06 | 51.21 | 0.53 | 32.65 | 83.11 | 92.87 |
| | CIL | 57.90 | 84.04 | 93.38 | 97.95 | **1.76** | **28.03** | **55.57** | **72.34** | **29.99** | **52.53** | **62.29** | **73.34** | **3.45** | **48.95** | **85.52** | **94.76** |
| MSMT17 | BoT | 9.91 | 48.34 | 73.53 | 85.29 | 0.07 | 5.28 | 20.20 | 31.11 | 2.75 | 15.78 | 25.92 | 35.50 | 0.09 | 16.10 | 59.06 | 76.48 |
| | AGW | 12.38 | 51.84 | 75.21 | 86.30 | 0.08 | 6.53 | 22.77 | 34.08 | 3.82 | 18.42 | 28.06 | 37.33 | 0.15 | 18.08 | 61.45 | 78.43 |
| | SBS | 10.26 | **56.62** | **82.02** | **90.39** | 0.05 | 7.89 | 28.77 | 40.00 | 3.23 | 22.71 | 36.68 | 46.53 | 0.12 | 21.16 | **70.65** | **83.95** |
| | CIL | **12.45** | 52.40 | 76.10 | 87.19 | **0.32** | **15.33** | **39.79** | **54.83** | **5.84** | **29.08** | **45.51** | **58.27** | **0.50** | **27.99** | 68.31 | 82.87 |
| CUHK03 | AGW | 49.97 | 62.25 | 64.64 | 81.50 | 0.46 | 3.45 | 5.90 | 11.59 | 12.69 | 17.20 | 16.26 | 26.29 | 2.89 | 19.40 | 33.43 | 53.85 |
| | CIL | **53.87** | **65.16** | **67.29** | **83.79** | **4.25** | **16.33** | **22.96** | **39.89** | **26.61** | **34.62** | **34.03** | **50.44** | **9.07** | **31.81** | **46.81** | **69.66** |

Table 5: Corruption invariant person ReID benchmarks on cross-modality datasets. For SYSU-MM01 dataset, Mode A and Mode B mean all-search (including indoor and outdoor cameras) and indoor-search experimental settings, respectively. For RegDB dataset, Mode A and Mode B represent visible-to-thermal and thermal-to-visible experimental settings, respectively. Note that we only corrupt RGB (visible) images in the corruption evaluation.

| Dataset | Method | Mode A | | | | | | | | Mode B | | | | | | | |
|---|---|---|---|---|---|---|---|---|---|---|---|---|---|---|---|---|---|
| | | Clean Eval. | | | | Corrupted Eval. | | | | Clean Eval. | | | | Corrupted Eval. | | | |
| | | mINP | mAP | R-1 | R-5 | mINP | mAP | R-1 | R-5 | mINP | mAP | R-1 | R-5 | mINP | mAP | R-1 | R-5 |
| SYSU-MM01 | AGW | 36.17 | **47.65** | **47.50** | **74.68** | 14.73 | 29.99 | 34.42 | 62.26 | **59.74** | **62.97** | **54.17** | **83.50** | 35.39 | 40.98 | 33.80 | 61.61 |
| | CIL | **38.15** | 47.64 | 45.41 | 73.95 | **22.48** | **35.92** | **36.95** | **65.54** | 57.41 | 60.45 | 50.98 | 81.34 | **43.11** | **48.65** | **40.73** | **71.44** |
| RegDB | AGW | 54.10 | 68.82 | **75.78** | **85.24** | 32.88 | 43.09 | 45.44 | 55.26 | 52.40 | 68.15 | **75.29** | 83.74 | 6.00 | 41.37 | **67.54** | 81.23 |
| | CIL | **55.68** | **69.75** | 74.96 | 84.71 | **38.66** | **49.76** | **52.25** | **65.83** | **55.50** | **69.21** | 74.95 | **86.12** | **11.94** | **47.90** | 67.17 | **83.25** |

CIL baseline using a two-stream architecture on the cross-modality visible-infrared ReID task. As seen by the results in Tab. 5, our baseline CIL considerably improves corruption robustness while compromising little performance on clean test sets.

We conduct ablation experiments on the components of our proposed baseline, as shown in Tab. 6. The standard ResNet-50 we use here is built on the AGW baseline, which deletes the non-local block and adds the loss function used by the SBS baseline. It can be seen from Table 6 that our

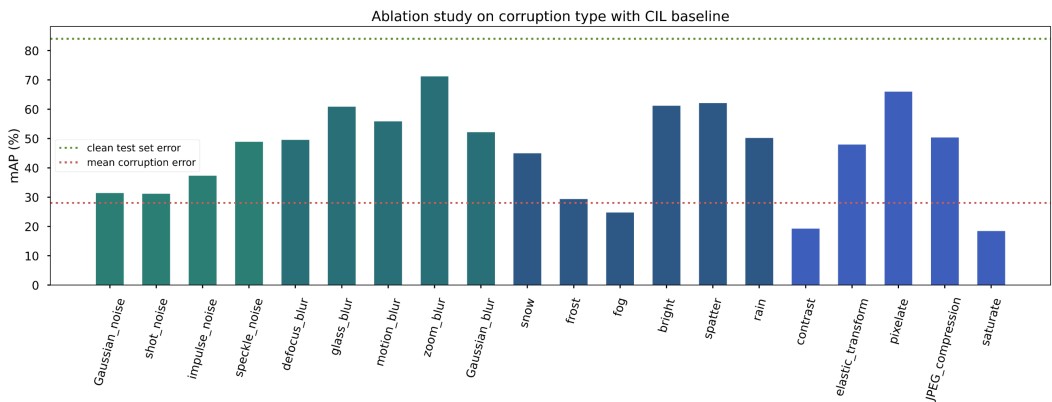

Figure 5: Ablation study on different corruption types (corrupted query and gallery), including 20 types of algorithmically generated corruptions from noise, blur, weather, and digital categories.

Table 6: Ablation study on CIL components, including pre-BNNeck inference, local-based augmentation (soft random erasing and self patch mixing) and consistent ID loss.

| Component | Clean Eval. | | | | Corrupted Eval. | | | | Corrupted Query | | | | Corrupted Gallery | | | |
|---|---|---|---|---|---|---|---|---|---|---|---|---|---|---|---|---|
| | mINP | mAP | R-1 | R-5 | mINP | mAP | R-1 | R-5 | mINP | mAP | R-1 | R-5 | mINP | mAP | R-1 | R-5 |
| Standard ResNet-50 | **60.57** | **85.52** | **94.48** | 97.95 | 0.55 | 15.26 | 39.87 | 55.84 | 21.67 | 38.36 | 45.00 | 55.29 | 1.27 | 38.48 | 83.50 | 93.48 |
| + infer. before BNNeck | 47.39 | 79.58 | 91.66 | 97.15 | 0.89 | 18.16 | 42.41 | 58.98 | 20.53 | 42.94 | 53.35 | 65.29 | 2.14 | 42.65 | 82.23 | 93.27 |
| + soft random erasing | 59.57 | 84.74 | 93.26 | **98.07** | 1.37 | 25.98 | 53.89 | 70.92 | 29.44 | 50.96 | 60.11 | 71.89 | 2.67 | 46.82 | 85.68 | 94.86 |
| + self patch mixing | 55.96 | 82.93 | 93.05 | 97.51 | **1.78** | 27.59 | **57.37** | 71.81 | **30.33** | **53.69** | **64.08** | **75.81** | 3.23 | 48.08 | 84.70 | 94.59 |
| + consistent ID loss | 57.90 | 84.04 | 93.38 | 97.95 | 1.76 | **28.03** | 55.57 | **72.34** | 29.99 | 52.53 | 62.29 | 73.34 | **3.45** | **48.95** | **85.52** | **94.76** |

suggested pre-BNNeck inference and local data augmentation approaches can increase the corruption robustness, and the consistency ID loss can effectively maintain the corruption robustness while boosting the performance on clean samples. In addition, we also perform ablation experiments of different corruption types on our CIL baseline to see the impacts of each individual corruption, as shown in Fig. 5. Experimental results are also averaged after ten evaluations. We can see that the model is more vulnerable to corruption types such as saturate, contrast, and fog that produce greater color interference.

## 5   Conclusion

In this work, we present detailed, large-scale robustness evaluations of 21 advanced ReID methods. Based on the study, we have some interesting findings and build a strong baseline on robust person ReID in the hope of extracting practical lessons for the broader community. First of all, we demonstrated the transformer's potential on ReID. Even when images are corrupted, they can still extract rich structured patterns. Moreover, given the limitations of the existing commonly used data augmentation, we design two new simple but effective data augmentation methods for mining more robust local features. Additionally, we discover that cross-dataset generalization increases with corruption robustness in ReID, which was overlooked by previous research on corruption robustness and may serve as an inspiration for generalizable person ReID.

**Limitations and broader impact.**   Compared with other baselines, the performance of our proposed baseline on clean samples is slightly degraded. Additionally, we cannot establish a clear relationship between corruption robustness and performance on clean images in ReID tasks. As for positive impact, we demonstrate through extensive experiments that synthetic corruption robustness contributes to performance on naturally occurring distribution shifts. Hence our synthetic corrupted datasets can serve as useful proxies and have the potential to mitigate ethical concerns associated with the collecting of vast volumes of pedestrian data.

However, when an object ReID system is used to identify pedestrians and vehicles in a surveillance system, it may infringe people's privacy. Because ReID system typically (not all) depends on unauthorized surveillance data, which means that not all human subjects were known they were being recorded. As a result, governments and officials must take considerable steps to develop stringent regulations and legislation governing the use of ReID technology. Otherwise, malicious agents may be able to monitor pedestrians or vehicles without their consent using multiple closed-circuit television cameras [9]. Additionally, researchers should avoid using datasets that raise ethical concerns. For instance, the DukeMTMC dataset [34] should no longer be used after it was shut down for violating data collection restrictions. Also, our benchmarks have excluded evaluation on DukeMTMC. Meanwhile, it is worth mentioning that the demographic composition of datasets does not accurately reflect the general population. Current data-driven deep learning systems only learn what is taught to them. Accuracy and fairness are jeopardized if they are not taught with diverse datasets. Each of us expects the ReID system to perform equally well across different individuals or populations. Therefore, the research community and developers need to be thoughtful about what data they use for training. This is essential for developing artificial intelligence systems which can help to make the world more fair [28].

**Acknowledgments and disclosure of funding.**   This work is supported by the National Natural Science Foundation of China under Grant No. 61972188 and No.62122035.

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
