# OpenReview forum: "Benchmarks for Corruption Invariant Person Re-identification"
_NeurIPS.cc/2021/Track/Datasets_and_Benchmarks/Round2 — NeurIPS 2021 Datasets and Benchmarks Track (Round 2)_

### Official Review · Reviewer_QWq5 · 2021-09-18
**Benchmarks for Corruption Invariant Person Re-identification**

**Rating:** 7
**Confidence:** 4
**Correctness:** Yes
**Clarity:** Yes, the paper is well written.

**Strengths:**

1. The benchmark is important for real-world applications on person reid, and the author presents an extensive comparison of current methods.
2. The connection between cross-domain generalization and corruption invariant seems very interesting, I wonder if that conclusion generalizes to other cross-domain tasks other than reid?
3. The proposed strong baseline for corruption invariant reid seems to work well.

**Weaknesses:**

1. The paper doesn't provide documentation about the licensing and maintenance plan.
2. My biggest concern is the ethical issues that may arise with the use of DukeMTMC dataset, The DukeMTMC dataset's author has already taken the DukeMTMC dataset offline because of ethical issues, https://www.dukechronicle.com/article/2019/06/duke-university-video-analysis-research-at-duke-carlo-tomasi https://freedom-to-tinker.com/2020/10/21/facial-recognition-datasets-are-being-widely-used-despite-being-taken-down-due-to-ethical-concerns-heres-how/, and the derivatives of DukeMTMC dataset is also removed from online e.g. https://github.com/liminxian/DukeMTMC-SI-Tracklet.
I don't know if other ReID datasets (Market 1501 etc.) have similar concerns, but I think this benchmark may enable unintended and unethical researches, so I vote for rejection.

**Additional Feedback:**

No

**Documentation:**

No

**Ethics:**

This paper uses the DukeMTMC dataset which have been taken offline by the original authors (https://www.dukechronicle.com/article/2019/06/duke-university-video-analysis-research-at-duke-carlo-tomasi), and I think this paper may enable unintended and unethical researches.

**Relation To Prior Work:**

Yes

**Summary And Contributions:**

This paper presents a new benchmark about corruptions of images in person re-identification datasets.
The paper directly uses the corruptions from ImageNet-C to create the corrupted test set for person reid with a newly added corruption subtype: rain.
The experiments made some observations that the random erasing augmentation, BNNeck helps under corruptions, and the paper established a strong baseline for corruption invariant person reid.
The paper also found that cross-domain generalization may be connected to the corruption invariance.

However, this paper doesn't include documentation for a licensing and maintenance plan, and the dataset about DukeMTMC (and IMO other ReID datasets) may have potential ethical issues, so I tend to vote for rejection at the current stage if the ethical issues are not properly resolved.

Post rebuttal:
I think the author response resolve my concerns about ethical issues, I raised my score to 7.

---

> ### Author Response · Authors · 2021-09-29
> **Response to Review #4**
>
> **Q1:** The paper doesn't provide documentation about the licensing and maintenance plans.
>
> **A1:**  To begin, we must declare that our work is a benchmark based on existing  datasets rather than a new dataset, which means that we are not required to offer documentation or maintenance plans. However, we appreciate the reviewers' suggestions and have added the license, more thorough documentation, and maintenance plan for our benchmark to the open-source URL.
>
> **Q2:** The biggest concern is the ethical issues that may arise with the use of DukeMTMC dataset.
>
> **A2:** We appreciate reviewer alerting us of the ethical concerns regarding the DukeMTMC dataset. The ethical concerns do exist and should not be ignored in person ReID community. To remedy this ethical issue, we make the following revisions and clarifications.
>
> 1. We have eliminated all the DukeMTMC related experiments in the updated version. Meanwhile, we have completed related experiments, and our findings remain unchanged.
> 2. We have added a discussion of ethical concerns about ReID datasets to the Broder Impact paragraph in the revised version. Besides, we have rechecked the other datasets we evaluated and found no similar ethical issues.
> 3. We neither create nor spread new real-world datasets. We just provide a technique for simulating real corrupted data. This technique can be applied to a variety of pedestrian datasets.
> 4. This work provides a possible solution to the ethical issue of person ReID datasets. Our technology has the potential to mitigate ethical concerns associated with the collecting of vast volumes of pedestrian data, as we have demonstrated through extensive experiments that synthetic corruption robustness contributes to performance on naturally occurring distribution shifts.

---

### Official Review · Reviewer_7yv1 · 2021-09-19
**Technical Review**

**Rating:** 6
**Confidence:** 3

**Strengths:**

S1 - The evaluation is extensive and covers a wide range of methods and datasets. Thus, the benchmark is valuable for the community.

S2 - The findings are interesting. Although it seems somewhat logical, that by design transformers will be more robust to some of the corruptions, the findings should lead to improved models in the future. Further, it is surprising that there seems to be no correlation between the performance on clean data and the performance on corrupted data.


**Weaknesses:**

Overall, there is not much technical innovation in this paper, as it uses the corruption strategy of [14] for the person re-identification task (plus a new rain corruption and a soft version of some augmentation strategies). However, since this is a benchmark, I do not see a strong necessity for technical novelty.

W1 - There are some statements that could be quantified numerically. For example, the paper claims that there is no obvious correlation between clean and corrupted performance. This could be actually measured with a correlation test so that this statement becomes stronger. The same applies for other statements such as the correlation between corruption performance and generalization.

W2 - The message of the paper gets slightly diluted by the introduction of modified augmentation strategies and adjusted loss terms. Ideally, a benchmark should only compare existing methods in new settings or under new criteria/data and not introduce novel methodologies. Here, the  problem is not very severe since the introduced modifications are in the same spirit as the benchmark, but a cleaner separation in terms of writing could be possible.

W3 - The heat maps in Fig2b and the supplement do not really show anything, as far as I can tell. For a proper comparison the visualization should be done on the same image. Even then, heatmaps are a very weak tool to show what a model has learned.


**Additional Feedback:**

The paper does not need to be anonymized for single blind reviewing.

**Clarity:**

The paper is clear in terms of writing and presentation of the findings of the benchmark.



**Correctness:**

I am not an expert in person re-identification. As far as I can tell, the findings and reasoning in the paper is correct and founded in experimental evidence.


**Documentation:**

As I understand the authors’ intent, the submission is a benchmark and not a dataset. Thus, no documentation was provided. A github link to the code to generate the image corruptions is given.

**Ethics:**

Person re-identification is a sensitive topic due to obvious and immediate applicability for surveillance and other ethically critical topics.
Yet, the paper does not contain any discussion of the ethical aspects of the topic nor does it disclose the funding source of the work.
While there is no new dataset introduced in this paper, the utilized existing datasets do contain several ethical issues, such as consent, privacy for the collected images.
For this reason, I would like to flag this submission for an additional ethics review.



**Relation To Prior Work:**

The relation to existing dataset and state of the art methods is sufficiently pointed out.



**Summary And Contributions:**

The paper is about the robustness of person re-identification methods with respect to image corruptions. The main contribution is an extensive study of 21 current methods on 6 datasets under the proposed corruption settings. The main findings are that a transformer based model retains significantly more performance under corruption than all other models, and that cross dataset generalization is correlated with corruption robustness.

---

> ### Author Response · Authors · 2021-09-29
> **Response to Review #3**
>
> **Q1:** Quantify correlation numerically.
>
> **A1:** We have updated the revised version to include quantitative correlation indicators (see lines 210-213). the cross-dataset generalization exhibits a consistent upward trend with the corruption robustness (correlation coefficient $\rho = 0.97$). However, there is no obvious correlation between cross-dataset generalization and clean sample performance (correlation coefficient $\rho = 0.22$).
>
> **Q2:**  Confusing for heat map.
>
> **A2:** We have improved the readability of Fig. 2 (b) in the revised version. The presented heat maps are generated by models trained with different augmentation techniques but fed the same image.
>
> **Q3:** Ethical issues and funding source
>
> **A3:** In the updated version, we have added a discussion of ethical concerns about the ReID datasets to the Broader Impact paragraph, as well as an Acknowledgement paragraph to disclose our funding source. This work is supported by the National Natural Science Foundation of China under Grant No. 61972188.

---

### Official Review · Reviewer_7YHU · 2021-09-21
**Interesting study but missing some key evaluations**

**Rating:** 6
**Confidence:** 5
**Correctness:** The claims are corrected.
**Clarity:** Overall it is well written.

**Strengths:**

1) This paper  is well motivated. In real world, data collected by surveillance cameras often suffer from various corruptions, e.g., occlusion, light change, insufficient light exposure, various weather conditions, etc.  Although existing methods have reached impressive performance on the clean datasets, they might not be satisfactory  when applied in practice. The paper aims to study this problem and provides a benchmark to systematically evaluate the re-ID performance under the more realistic settings.

2)  The benchmark is comprehensive. It includes 6 datasets with both single and multiple modalities, 20 types of corruptions commonly occurred in real world, and 5 levels of severity.

3) The experiments are extensive. Up  to 21 recent re-ID methods are employed to be evaluated on the benchmark.

4) Some interesting observations have been reached, which could inspire future developments for this topic.

5) A strong baseline has been proposed which reaches competitive performance for different experimental settings.



**Weaknesses:**

1) When evaluating methods on the corrupted datasets, it is unclear whether these methods are trained on corrupted data.  If the methods are trained with corruptions, it is surprising that the performance is so low. This is because if the models is trained to be robust with various corruptions, they should perform reasonably well when applied on the test data.

2) Following the above, if the methods are trained with corrupted data, how do you execute the training processing? What is the sampling strategy and how do you apply different corruptions? Are the training processing the same for all the compared methods?

3) All the corruptions are mixed together; it is unclear to see the impacts of each individual corruption. This is interesting because in practice we usually know the types corruptions that are more likely to be suffered considering that mounting location of the surveillance cameras. For example, indoor cameras are unlikely to have raining and foggy problems.

4) The ablation study of the proposed baseline should be included in the main text, rather than in the supplementary material. They are essential to evaluate the contributing role of each components.

**Additional Feedback:**

See the above.

**Documentation:**

The documentation is good.

**Ethics:**

No ethical concerns.

**Relation To Prior Work:**

Relation to prior work has been clearly presented.

**Summary And Contributions:**

This paper proposes a new benchmark for studying the performance of re-ID models on corrupted person images. The benchmark builds on top of several most commonly used re-ID datasets and applies various corruption transformations on the datasets. With the benchmark, extensive experiments are conducted and some conclusions have been drawn. Moreover, a strong baseline is proposed which reaches competitive performance for both single and cross-modality re-ID experiments.

---

> ### Author Response · Authors · 2021-09-29
> **Response to Review #2**
>
> **Q1:** Model trained with corrupted data?
>
> **A1:** No. The benchmark datasets are constructed based on agnostic corruption types that are not encountered in model training. Since corruption types in the real world are numerous and unpredictable, it will be more practical to learn a corruption generalized model without additional training and adaptation.
>
> **Q2:** Unclear to see the impacts of each individual corruption.
>
> **A2:** We have added ablation experiments for different corruption types in the revised version. The experiments are conducted on our CIL baseline and the results are averaged after ten evaluations. What we can conclude is that the model is more vulnerable to corruption types such as saturate, contrast, and fog that produce greater color interference.
>
> **Q3:** Baseline ablation study should be included in the main text.
>
> **A3:** We have added ablation experiments for different components of the baseline in the revised version.

---

### Official Review · Reviewer_Fm5X · 2021-09-21
**Review for Paper 49**

**Rating:** 7
**Confidence:** 4

**Strengths:**

* Experiments are extensive and major recent ReID models have been evaluated on standard ReID benchmark datasets.
* Study is well motivated and addresses important aspect for real-world applications of ReID models
* Makes proper connections and references to previous findings by relevant prior work
* In addition to corruption benchmark, they also study the effect of data augmentation and cross-dataset performance

**Weaknesses:**

See comments below in correctness and clarity sections. Please have these concerns addressed in revision.

**Additional Feedback:**

n/a

**Clarity:**

Generally, paper reads well. However, clarity on experiment could be improved.

It would be good to specify somewhere in the paper what kind of the augmentation and evaluation set (corrupt eval or query or gallery) were used for the corrupt input performance reported in Figs 1,3,4.

Figure 2 - (b) not very clear or convincing that model trained with soft random erasing and random patch mixing capture more discriminative parts. Also readability of the figure could be improved by writing augmentation method used for each activation map instead of putting it in the caption.

**Correctness:**

Authors mention "Corrupted images can be seen as heterogeneous data for clean images, and our corruption invariant benchmarks as a setting in the open world". However, open world setting specifically mean the setting where the identities differ between query and gallery and this statement is misleading.

The authors claim and experimentally show that soft random erasing is doing less harm than hard random erasing. However, according to Figure 4, soft random erasing also consistently seem to harm corrupt robustness as erasing degree increases. What is the main motivation to use erasing augmentation to begin with? Do it make model more robust compared to 0 erasing rate?

**Documentation:**

Error bars are not reported in the main paper. I would suggest to put them in main text along with figures.

**Ethics:**

n/a - benchmark paper

**Relation To Prior Work:**

The paper make good connections to many prior works.

**Summary And Contributions:**

This paper studies the robustness of ReID models under a diverse range of image corruptions. The type of corruptions considered in the paper follow the work of [14] with a few additional augmentation. By benchmarking 21 recent ReID models on 6 popular ReID datasets, the study reveals that majority of recent ReID models did not make much progress on corruption robustness. Based on their experiments, the authors propose a strong baseline that is more robust to input corruption compared to other baseline models trained with popular practices.

---

> ### Author Response · Authors · 2021-09-29
> **Response to Review #1**
>
> **Q1:** About open world setting.
>
> **A1:** We have different definitions of open world. The setting where the identities differ between query and gallery is defined as the open-set ReID setting in our work. As for the concept of open world, we follow the definition provided in [1]. The open world setting includes heterogeneous Re-ID by matching person images across heterogeneous modalities, end-to-end Re-ID from the raw images/videos, semi-/unsupervised learning with limited/unavailable annotated labels, robust Re-ID model learning with noisy annotations, and open-set person Re-ID when the correct match does not occur in the gallery [1]. Thank you very much for your kind suggestion about the confusing concept we used. We have made clearer expressions in the revised version.
>
> [1] Ye, Mang, et al. "Deep learning for person re-identification: A survey and outlook." IEEE Transactions on Pattern Analysis and Machine Intelligence (2021).
>
> **Q2:** About soft random erasing.
>
> **A2:** It is true that the soft random erasing will hurt corruption robustness as erasing probability increases. The random erasing is a widely used and effective data augmentation technique for ReID tasks. But in our work, we found that hard random erasing helps with performance on clean samples but hurts corruption robustness when compared to not employing erasing augmentation. The soft random erasing is proposed to alleviate this issue. Under the same erasure probability, the soft random erasure provides better corruption robustness (especially 0.5). In the revised version, we have refined our language to more accurately convey our findings.
>
> **Q3:** About clarity.
>
> **A3:** In the revised version, we have added the specific evaluation sets used in Fig1, 3, and 4, and mentioned our default data augmentation settings in the Benchmark section (see lines 136-138). Besides, we have improved the readability of Fig. 2 (b) and added the related error bar.

---

### Author Response · Authors · 2021-09-30
**General Response**

We thank reviewers for their insightful comments. We were encouraged that they found our work
- is well motivated and addresses important aspect for real-world applications of ReID models; **(R1)**
- makes some interesting observations, which could inspire future developments for this topic; **(R2)**
- covers a wide range of methods and datasets and valuable for the community. **(R3)**


In the revised version, we have addressed the following concerns:
1.	Ethical issue arise with the use of DukeMTMC dataset **(R3, R4)**
- We have eliminated all the DukeMTMC related experiments in the updated version. Meanwhile, we have completed related experiments on the other dataset, and our findings remain unchanged.
- We have added a discussion of ethical concerns about ReID datasets to the Broder Impact paragraph in the revised version. Besides, we found no similar ethical issues to the other datasets we evaluated. Referencing the related ethical concerns [1] happened in NeurIPS 2020, we have followed the officially recognized solution [1] (see link: https://papers.nips.cc/paper/2020/file/821fa74b50ba3f7cba1e6c53e8fa6845-MetaReview.html) to revise our work to avoid ethical issues.
2.	Not quantify correlation numerically **(R3)**
- We have updated the revised version to include quantitative correlation indicators.
3.	Figure 2 (b) is not very clear **(R1, R3)**
- We have improved the readability of Fig. 2 (b) in the revised version.
4.	Unclear to see the impacts of each individual corruption. Lack of baseline ablation study in the main text. **(R2)**
- We do have conducted the mentioned experiments but presented the results in the Appendix. In the revised version, we have added ablation experiments for different corruption types and for different components of the baseline **from the Appendix to the main text**)
5.	Some statements are not clear **(R1)**
- In the revised version, we have refined our language to more accurately convey our findings.


**Besides,**
we appreciate **R4** alerting us of the ethical concerns regarding the DukeMTMC dataset. We quite agree that the ethical issues do exist and should not be ignored, so we have eliminated all the DukeMTMC related experiments in the updated version. ~~However, we argue that it is unfair to maintain the decision (rejection) after addressing the **R4**’s biggest concern.~~ We **do not** think the updated version of this paper will enable unintended and unethical researches.
- First, we have eliminated all the DukeMTMC related experiments in the updated version.
- Second, we neither create new real-world datasets nor spread any existing datasets. We just provide a technique for simulating real corrupted data.
- Finally, this work provides a possible solution to the ethical issue of person ReID datasets. We demonstrate through extensive experiments that synthetic corruption robustness contributes to performance on naturally occurring distribution shifts. Hence our synthetic corrupted datasets can serve as useful proxies and have the potential to mitigate ethical concerns associated with the collecting of vast volumes of pedestrian data.

[1] Ge, Yixiao, et al. "Self-paced contrastive learning with hybrid memory for domain adaptive object re-id." arXiv preprint arXiv:2006.02713 (2020).

---

### Comment · Program_Chairs · 2021-10-13
**Official Ethics Review**

It seems the authors have addressed the following issues in the “Limitations and broader impact” section of their paper:
- Use of the depreciated Duke MTMC dataset
- Acknowledgement for potential for surveillance use
- Acknowledgement of privacy and consent violations in data collection regarding identification objectives
- Acknowledgement of demographic representation issues with large human-centered datasets

They should add citations and further elaborate on the risks of demographic homogeneity in these datasets [1] but otherwise that section adequately covers the topics we hoped to see them mention explicitly. If relevant, authors should also consider any potential data distribution or copyright limitations that may interfere with their ability to make use of the dataset and should be conscientious in disseminating their models responsibly.

[1] Merler, Michele, et al. "Diversity in faces." arXiv preprint arXiv:1901.10436 (2019).

---

> ### Author Response · Authors · 2021-10-15
> **Response to Official Ethics Review**
>
> Thanks a lot for your careful review and valuable suggestions. We have added citations and further elaborated on the risks of demographic homogeneity in re-ID datasets (see lines 289-294).
>
> The datasets included in our benchmark are all officially allowed for the use of non-commercial research. These datasets can be obtained via official download links or by contacting associated authors.
>
> We will disseminate models carefully and responsibly. We will only distribute our models to researchers who have signed an agreement and pledged to utilize the models solely for scientific research purposes. We have included our detailed contact information in the open-source URL to facilitate researchers requesting our models.

---

### Decision · Program_Chairs · 2021-10-09

**Decision:**

Accept

**Comment:**

This paper studies the performance of re-ID models on corrupted person images with a new benchmark. Generally, all reviewers acknowledge the motivation and importance of the study, and find it comprehensive enough to serve as a good benchmark paper. Apart from some correctable writing and clarity issues, a critical concern raised by a reviewer is the ethical issue of containing a previous dataset (DukeMTMC) which is already retracted. The authors properly addressed this issue by omitting DukeMTMC from the work. The reviewer was satisfied with this action and raised the score to acceptance.
Now the paper is supported by all reviewers, thus I recommend acceptance.

Flagged for an additional ethics review because this is a study on person reidentification.